# Exploring Sparse Adapters for Scalable Merging of Parameter Efficient Experts

**Samin Yeasar Arnob** [*]
McGill University, Mila, Microsoft

**Zhan Su**
Université de Montréal

**Minseon Kim**
Microsoft

**Oleksiy Ostapenko**
ServiceNow

**Doina Precup**
McGill University, Mila

**Lucas Page-Caccia**
Microsoft

**Alessandro Sordoni**
Microsoft

## Abstract

Model merging aims to integrate knowledge from multiple finetuned experts into a single, unified multi-task model. To Merging parameter-efficient task experts has recently gained growing attention as a way to build modular architectures that can be rapidly adapted on the fly for specific downstream tasks, without requiring additional fine-tuning. Typically, LoRA serves as the foundational building block of such parameter-efficient modular architectures, leveraging low-rank weight structures to reduce the number of trainable parameters. In this paper, we study the properties of sparse adapters, which train only a subset of weights in the base neural network, as potential building blocks of modular architectures. First, we propose a simple method for training highly effective sparse adapters, which is conceptually simpler than existing methods in the literature and surprisingly outperforms both LoRA and full fine-tuning in our setting. Next, we investigate the merging properties of these sparse adapters by merging adapters for up to 20 natural language processing tasks, thus scaling beyond what is usually studied in the literature. Our findings demonstrate that sparse adapters yield superior in-distribution performance post-merging compared to LoRA or full model merging. Achieving strong held-out performance remains a challenge for all methods considered.

## 1 Introduction

Multitask training, e.g. (Raffel et al., 2019), is an effective method to improve the performance of large language models (LLMs) across different tasks. However, for multitask training, all task-specific datasets need to be available simultaneously requiring data sharing during training. Model merging has emerged as an efficient alternative to building multi-task models (Wortsman et al., 2022), which allows tasks to be trained separately and then combined at the end of the training process, thus ensuring privacy of data and savings at training time. Recent work shows that model averaging can improve out-of-distribution performance over multitask training (Yadav et al., 2024b; Ostapenko et al., 2024). However, merging models do not achieve the same performance as true multitask training due to weight interference, and requires careful weight manipulation (Ilharco et al., 2023; Yadav et al., 2023; Akiba et al., 2024; White, 2016; Davar, 2024) during the merging process to resolve conflicts.

Merging has recently been the focus of modular architectures that re-use parameter-efficient experts such as LoRA (Hu et al., 2021b) readily available on platforms such as Huggingface Hub (Huang et al., 2024; Ostapenko et al., 2024; Muqeeth et al., 2024). Recent evidence suggests that carefully composing LoRA (Hu et al., 2021a) modules can even outperform multi-task training on some tasks (Prabhakar et al., 2024). The widespread adoption of LoRA is due to the fact that it reduces the number of task-specific trainable parameters via low-rank decomposition while still maintaining good task performance. However, merging task-specific LoRA experts might result in significant parameter interference given that all parameters of a given layer are modified for each task. In contrast, sparse fine-tuning methods – alternative parameter-efficient fine-tuning methods that train a

---

[*]Corresponding author: `samin.arnob@mail.mcgill.ca`

smaller sub-network within the base model for each task, have been proposed (Ansell et al., 2024; 2021b; Frankle & Carbin, 2019) and have shown some promising results in composability (Ansell et al., 2021b), albeit in a constrained setting of merging a language expert with a task expert for multi-lingual tasks.

In this paper, we study the properties of sparse adapters as potential building blocks for modular architectures. We begin by introducing a simple yet straightforward method for training sparse adapters, which simplifies prior approaches while delivering superior performance compared to LoRA and full fine-tuning. We leverage *connection-sensitivity* (Mozer & Smolensky, 1988; Lee et al., 2018) to identify an important subset of parameters important to the task. We also propose a more structured sparse representation, *block-sparse*, which is particularly advantageous due to the efficient use of CUDA kernels (Yamaguchi & Busato, 2021; Gray et al., 2017; Su et al., 2024). Next, we investigate the merging properties of sparse adapters by conducting experiments across a set of 20 FLAN Longpre et al. (2023) tasks, expanding the typical scope of sparse adapter merging, which has been limited to few tasks in prior literature (Ansell et al., 2021a; Panda et al., 2024). We compare the performance of sparse adapters with LoRA and full fine-tuning. We evaluate performance on both the test sets of the 20 held-in tasks (held-in) and a set of unseen 10 tasks (held-out). Additionally, we benchmark recent merging methods, including Task Arithmetic (Ilharco et al., 2023), Ties (Yadav et al., 2023) and Breadcrumbs (Davar, 2024).

Our results show that sparse adapters outperform both LoRA and full fine-tuning in a single fine-tuning experiment across 20 tasks. Moreover, merging sparse adapters retains strong held-in performance while maintaining competitive held-out results. Unlike full-finetuning merging methods, which degrade in performance when scaled to 20 experts, sparse-adapters prove to be more effective. Through ablation studies, we identify that the degradation of held-in performance is more due to parameter modifications outside of the sparse masks, rather than interference within the masks themselves. Overall, our work demonstrates that sparse adapters offer a scalable and efficient approach to merging modular architectures for multitask learning, especially when extending beyond the typical two-task setting explored in previous research. While we show improved generalization for held-out tasks compared to multitask training, a notable gap remains for held-in performance.

## 2 RELATED WORK

**Parameter-Efficient finetuning (PEFT)** enable the efficient adaptation of LLMs through updating only a small subset of parameters (Han et al.). PEFT approaches directly update the pre-trained weights in a parameter-efficient manner (Hu et al., 2021a; Zhang et al., 2023; Hayou et al.; Liu et al., 2024; Dettmers et al., 2024). The most prominent method is Low-Rank Adaptation (LoRA) (Hu et al., 2021b), which parameterizes incremental weight updates $\Delta$ is the product of two low-rank matrices. LoRA achieves performance comparable to or even surpassing that of full fine-tuning. More recently (Hu et al., 2025) introduced a follow-up LoRS method, that achieves better computing and memory efficiency for fine-tuning sparse LLMs.

Parameter-sparse training has become increasingly popular in deep learning for achieving results similar to dense training (Frankle & Carbin, 2019; Lee et al., 2018; Wang et al., 2020; Evci et al., 2021; Arnob et al., 2021; 2025). Recent research on training sparse networks for LLMs has mainly concentrated on single-task training and merging with a limited number of tasks. (He et al., 2022) evaluates the performance of various sparse training techniques for LLMs in a single-task context. Ansell et al. (2021a); Panda et al. (2024) investigates iterative magnitude pruning (Frankle & Carbin, 2019), Ansell et al. (2024) introduces an evolution-based sparse training approach that adopts the prune-and-grow method described in (Evci et al., 2021). Ansell et al. (2021a); Panda et al. (2024) shows that sparsity can help prevent catastrophic forgetting when model merging for Task A and Task B. In this work, we extend sparse adapters merging to the scale of 20 tasks, fine-tuning experts asynchronously on the FLAN dataset (Longpre et al., 2023) and evaluating performance with various merging methods.

**Expert Merging** There is growing interest in aggregating adapters from diverse domains through model merging techniques (Yadav et al., 2024a). The simplest form of merging involves averaging the weights of different experts. Expanding on weight averaging, Task Arithmetic (Ilharco et al., 2023) involving the creation and combination of task vectors facilitated multi-task learning. Beyond simple averaging, Yadav et al. (2023) propose TIES, and Akiba et al. (2024) introduce DARE, both

of which reset redundant parameters, resolve sign conflicts, and selectively merge parameters that demonstrate sign consistency. Similarly, Davar (2024) propose Breadcrumbs, a method that eliminates weight outliers and filters out negligible perturbations. Some methods like Fisher Merging (Matena & Raffel, 2022a) and RegMean (Jin et al., 2023b) need training data-based pre-computations to measure individual parameter importance but these are highly memory and data intensive. All of these merging methods require tuning merging hyper-parameters and carefully updating weights to avoid conflicts during model merging. We demonstrate that sparse adapters can be merged using simple weight averaging, yielding the best performance on held-in datasets while maintaining competitive generalization on held-out tasks.

## 3 TRAINING AND MERGING SPARSE ADAPTERS

We learn a sparse adapter for a task, where the task-dependent shift to the base model weights $\Delta W$ is sparse, $\Delta W = \hat{W} \cdot M$, where $M$ has entries in $\{0, 1\}$ and $\hat{W}$ is a dense weight. We initialize the trainable parameters $\hat{W}$ as zeros and learn the task specific mask $M$ using a parameter selection criteria that we present next. We propose two Sparse Adapters: **(a)** Element-sparse, where each parameter in a linear layer is scored individually, and **(b)** Block-sparse adapter, where we consider non-overlapping blocks of size $B$ within a linear layer. To ensure scalability when training large language models, we focus on training only the Query-Key-Value ($QKV$) layers.

---

**Algorithm 1** Sparse Adapter Training

**Init:** # Base params, Trainable params, Sparse Mask
  ▷ $W \in \mathbb{R}^{d_1 \times d_2}$, $\hat{W} = \mathbf{0}^{d_1 \times d_2}$, $M = \mathbf{1}^{d_1 \times d_2}$
  ▷ Training dataset: $D_{\mathcal{T}}$, optimizer($\hat{W}$)
**Training loop:**
**for** epoch = 1 to 5 **do**
    **for** step = 1 to N **do**
        batch $\sim \mathcal{D}_{\mathcal{T}}$
        loss = model(batch, $W + \hat{W} \cdot M$)
        loss.backward()
        **if** epoch == 1 and step % 100 == 0 **then**
            ▷ $M = \text{Top}_K(|\hat{W} \cdot \hat{W}.\text{grad}|)$ # Eq. 1.
        **end if**
        optimizer.step()
    **end for**
**end for**

---

**Parameter selection criteria**   Our parameter selection criteria is motivated by the saliency based score proposed by Mozer & Smolensky (1988) and later shown to be effective for deep learning (Lee et al., 2018; Arnob et al., 2021; 2025) as a method for ranking parameters by their importance at initialization. We infer the task specific mask using saliency based score, $\mathcal{S}$ which measures the importance of every parameter in the neural network for a given task:

$$M = \text{Top}_k\Big(\mathcal{S}(\hat{W}; D_{\mathcal{T}})\Big), \tag{1}$$

where $D_{\mathcal{T}}$ is the dataset of the current task $\mathcal{T}$, $\text{Top}_k$ operator selects top-$k$ parameters and sets their corresponding mask values to 1 while the rest is set to 0.

For the criteria $\mathcal{S}$, we compute the influence of a specific parameter value on the loss function (Mozer & Smolensky, 1988; Lee et al., 2018). Formally, the effect of a given parameter $w_q \in \mathbb{R}$ on the loss is:

$$\mathcal{S}(w_q) = \lim_{\epsilon \to 0} \left| \frac{\mathcal{L}(W) - \mathcal{L}(W + \epsilon \delta_q)}{\epsilon} \right| = \left| w_q \frac{\partial \mathcal{L}}{\partial w_q} \right|, \tag{2}$$

where $\delta_q$ is a vector whose $q$-th element equals $w_q$ and all other elements are 0. The mask $M$ retains parameters based on the saliency scores above a given threshold. We use keep-ratio ($Kr$) to represent model sparsity, where $Kr$ denotes the proportion of parameters that are trainable. For instance, a 95% sparse model corresponds to a $Kr$ value of 0.05, meaning only 5% of the parameters are kept trainable. Accordingly, we set $k$ in Equation 1 as $Kr \times (d_1 \times d_2)$.

**Block sparsity** Instead of calculating importance per-weight, we also propose to use block sparsity, which identifies and retains important regions within a linear layer. Given a keep-ratio value $Kr$ and a linear layer of size $W_d \times W_k$, the number of blocks is calculated as:

$$N_B = Kr * \frac{(d_1 \times d_2)}{B \times B}, \tag{3}$$

where $B$ denotes the block size. We compute the importance score $\mathcal{S}$ for each block and mask the parameters within the top $N_B$ blocks. We refer to block-wise sparse training as Block-Sparse.

**Algorithm**

See Algorithm 1. We initialize $\hat{W} = 0$ and $M = 1$. During the first epoch of fine-tuning, both $\hat{W}$ and $M$ are updated. The mask $M$ is recalculated using Eq 1 every 100 gradient steps: we compute Equation equation 2 after the backward pass to derive the importance score for each weight. We then select the top-$k$ parameters based on these scores to construct an updated mask $M$. After the first epoch, the mask is kept fixed, and only the masked weights are fine-tuned. The overall algorithm is in Algorithm 1. The iterative update of $M$ outperforms single-shot update (see Figure 4).

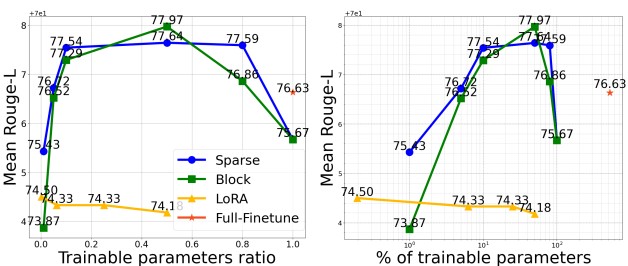

Figure 1: Comparison of sparse adapter, LoRA, and Full Fine-Tuning on **Single-Task** Performance. We report the average rouge-L performance across 20 tasks. We present performance variations across different ranges of trainable parameters, adjusting the tuning rank for LoRA and the fraction of parameters for sparse adapter applied over only the QKV layers. Here we compare (*Left:*) sparse adapter, LoRA and FFT (*Right:*) a more accurate comparison **in terms of numbers of trainable parameters.**

**Memory Requirements** Our approach is comparable to full fine-tuning of QKV layers until the first mask update step (at 100 gradient steps). After this occurs, we only keep on GPU the selected sparse parameters. Given that the mask is susceptible to change in the first epoch, we keep the parameters excluded from the current selection on CPU, in the case some of them might be selected by the successive mask updates. This strategy optimizes memory usage during training. After completing the first epoch, $M$ is fixed, and we can discard the unmasked weights.

**Comparison with Prior Sparse Fine-tuning** Our method is close to previous approaches. Ansell et al. (2021b); Panda et al. (2024) where a full fine-tuning for at least one epoch is needed to identify the sparse mask $M$. We show that updating the mask multiple times during the first epoch is useful. Ansell et al. (2024) offers a memory-efficient solution by maintaining sparsity throughout the training via a grow and prune method. We observe that we can maintain sparsity throughout the training with exception of the first 100 gradient updates. Given that we only we fine-tune only the $QKV$ layers, memory requirements do not increase significantly. Regarding parameter selection criterion, Ansell et al. (2021a) use the lottery ticket (Frankle & Carbin, 2019) criterion, while we apply a saliency-based criterion; a comparison between these criteria would be useful in the future.

**Merging Sparse Adapters** For each task $\mathcal{T}_i$, $i = \{1, \ldots, N\}$ we have a sparse task-specific shift $\Delta W_i = \hat{W}_i \cdot M_i$. To perform the adapter merging across tasks, we need to properly account for the fact that some individual weights might be trained by two or more tasks. In particular, if a weight element is shared across $k$ different task-specific masks, we need to average the updates for that weight element by dividing the sum by $k$. We compute a weight overlapping factor $F_o$, which reflects how many tasks have selected a particular weight. The merged update for the weights is:

$$\Delta W_m = \frac{1}{F_o} \sum_{i=1}^{N} \Delta W_i, \tag{4}$$

where $F_o = \min(\sum_{i=1}^{N} M_i, \mathbf{1})$, is the element-wise sum of the masks $M_i$ and $\sum_{i=1}^{N} M_i$ represents the count of how many tasks selected each weight element. We capped at 1 to prevent division by zero. Finally, the merged model weights are obtained by adding the merged sparse update $\Delta W_m$ to the base model weights: $W = W + \Delta W_m$.

## 4 EXPERIMENTS

We begin by evaluating the performance of sparse adapters in comparison to LoRA and full fine-tuning in Section 4.1. Following this, we explore various merging techniques applied to sparse adapters in Section 4.2.

**Setup** We choose the learning rate based on the hyper-parameter sweep for different methods (see Figure 4). We conduct our experiments using the FLAN dataset (Longpre et al., 2023) and sample 20 held-in and 10 held-out tasks following (Ostapenko et al., 2024), where each task is sub-sampled to 10,000 examples. Within these samples, 1,000 are allocated for validation and early stopping. For parameter-efficient fine-tuning, we load the base model $W$ in bfloat16 format and trainable parameters $\hat{W}$ in float32. As the base model, we use Phi-3-mini-4k-instruct (3.8B parameters) (Abdin et al., 2024) for all our finetuning experiments. For each FLAN task, we

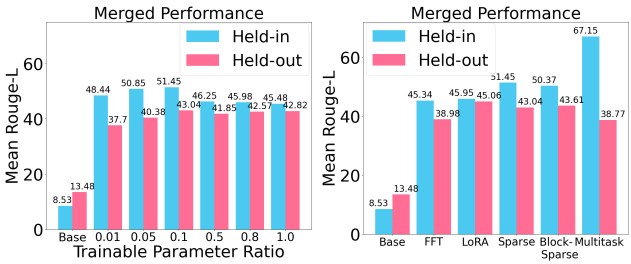

Figure 2: Performance of sparse adapter with Uniform Merging on held-in (20 tasks) and held-out (10 tasks) datasets. (*Left:*) shows the impact of varying the $Kr$ (fraction of trainable parameters) in sparse training performance. We compare the performance of the sparse adapter with the base model to demonstrate the improvement in fine-tuning. (*Right:*) We compare the performance with LoRA, FFT and Multitask training.

finetune the model for 5 epochs. For the hyperparameter sweep, we randomly select 5 tasks from the 20 held-in tasks and keep them fixed.

## 4.1 SINGLE TASK FINETUNING

For single-task performance comparison, we finetune 20 FLAN tasks for 5 epochs and provide the mean Rouge-L performance. In Figure 1, we compare the performance of sparse across different sparsity levels, alongside LoRA (Hu et al., 2021b) and full fine-tuning (FFT). Specifically, we adjust the sparsity of by varying the parameter $kr$, which represents the percentage of parameters retained during training. For instance, $kr = 0.01$ corresponds to a sparsity of 99%. In our experiments, both sparse and LoRA train the QKV layers. To ensure a fair comparison, we also adjust the rank of LoRA accordingly, maintaining consistency across the evaluation.

Our results show that sparse adapters with $kr = 0.01$ (99% sparsity) outperforms the fully fine-tuned model. Performance improves as $kr$ increases up to 0.5, after which we observe a gradual decline in performance at $kr = 0.8$. Interestingly, when $kr = 1$, which corresponds to dense training of the QKV layers, performance drops further. This suggests that training the QKV layers alone, even when fully dense, is not sufficient for optimal performance. Instead, selecting an appropriate subspace of parameters is crucial to achieving better results.

## 4.2 MODEL MERGING PERFORMANCE

We explore the merging of 20 expert models and compare their performance to multitask training. Fine-tuning a base model on specialized tasks can often result in a loss of generalization ability (Wortsman et al., 2022). As a result, we investigate whether combining the expertise of multiple sparse adapters can improve performance on held-out data over the multitask performance. We select 10 tasks from the FLAN dataset to evaluate the held-out performance of the models.

For model merging, we employ simple uniform weight averaging (Wortsman et al., 2022) while we benchmark other merging methods next. For sparse merging, we adopt our weighted averaging accounting for overlap between sparse models 4. We compare the merging performance of full-finetuning (FFT) and LoRA as baseline models. In the case of full-finetuning, we average the weights of multiple models equally. For LoRA, we compute the average over the low-rank adapters, by averaging $A$ and $B$ separately: $A_m = \sum_{i=1}^{N} A_i$ and $B_m = \sum_{i=1}^{N} B_i$.

**Performance under Varying Sparsity:** Figure 2(a) shows the performance improvement of the sparse adapter method over the Phi-3 base model at different $Kr$ values. We find that $Kr = 0.1$ provides the best merging performance for both held-in and held-out datasets. Although the best single-task performance without merging is achieved at a $Kr = 0.5$ (Figure 1), the increased weight population leads to greater interaction between weights, causing weight corruption that negatively impacts merging performance. This observation demonstrates a parameter saturation effect: as

| Method | Merging | % param trainable | Hparam | Mean Rouge-L | |
| --- | --- | --- | --- | --- | --- |
| | | | | Held-In | Held-Out |
| **Full-Finetune** | Averaging | 100% | | 45.33 | 38.98 |
| | Task-Arithmetic | 100% | ✔ | 39.40 | 25.31 |
| | Ties | 100% | ✔ | 50.21 | **46.16** |
| | Breadcrumbs | 100% | ✔ | 39.44 | 25.33 |
| **LoRA** | Averaging | 1.54% | | 45.96 | 44.01 |
| **Sparse** | Averaging | 2.37% | | **51.44** | 43.09 |
| **Block-Sparse** | Averaging | 2.37% | | 50.37 | 43.61 |
| **Multitask** | - | 100% | | 67.15 | 38.77 |

Table 1: Comparison of merging performance between full fine-tuning and PEFT methods across various merging techniques. We report the Rouge-L score for all approaches.

the number of parameters increases, the learning complexity of sparse training grows, leading to improved merging performance up to $Kr = 0.1$. However, beyond this point, more weight conflicts arise, leading to performance degradation when $Kr$ exceeds 0.1.

**Performance Compared to Multitask training** In Figure 2(b), we compare the performance of uniformly merged FFT, LoRA (Rank=128), and sparse adapter ($Kr = 0.1$) with multitask training. For held-in tasks, multitask training outperforms all merged PEFT methods, while sparse adapter shows the best model merging performance. However, for out-of-distribution generalization (held-out), LoRA ($> 13.52\%$) and sparse adapter ($> 11.01\%$ for elementwise-sparse, $> 12.48\%$ for block-sparse) significantly outperform multitask training.

**Performance Compared to Different Merging Methods** Various model merging methods typically fine-tune a pre-trained base model and compute a *task vector* (Ilharco et al., 2023) by subtracting the original model weights from those after fine-tuning on a specific task: $\tau^n = W_{\text{finetune}}^n - W$. These task vectors $\{\tau\}_{n=1}^N$ are then used to adjust the behaviour of the merged model. One straightforward approach, *Task-Arithmetic* (Ilharco et al., 2023), sums the task vectors and computes a weighted merge with the base model: $W_{\text{new}} = W + \lambda \sum_{n=1}^N \tau^n$. To address parameter interference caused by different task vectors, methods such as *TIES* (Yadav et al., 2023) trim less impactful task vectors by setting them to zero, resolving sign conflicts through majority voting among the vectors. *Breadcrumbs* (Davar, 2024) proposes filtering out outliers and removing negligible perturbations from the task vectors to improve merging performance. We also compare *Uniform weight-averaging* (Wortsman et al., 2022), which involves averaging the weights of multiple models fine-tuned on different tasks uniformly. We leave out computationally expensive approaches, such as those involving Fisher matrices (Matena & Raffel, 2022b), backward passes (Yang et al., 2024), or computing model activations (Jin et al., 2023a), as these methods do not scale well with large models or a high number of expert.

Despite employing simple weight averaging, sparse adapters achieve the highest Rouge-L scores (sparse: 51.44, block-sparse: 50.37) compared to other merging methods across 20 held-in tasks. Although sparse adapters utilize only 2.37% of the trainable parameters in comparison to FFT, they surpass the FFT-averaging by 13.48% and 11.12% in performance. Weight interpolation between pre-trained and fine-tuned models has been shown to improve out-of-distribution performance (Wortsman et al., 2021). Consistent with recent work (Yadav et al., 2024b), we find that most model merging methods outperform multitask training in terms of held-out performance. The mean Rouge-L scores across 10 held-out tasks are compared in Table 1, demonstrating that most of the model merging methods outperform multitask training in terms of generalization to unseen tasks. Notably, Ties achieves the best performance, while both sparse adapter and LoRA with simple averaging demonstrate competitive results.

## 5    CONCLUSION

In this paper, we explore the potential of sparse adapters as efficient building blocks for modular architectures in multitask learning. Our proposed method for training sparse adapters is conceptually

simpler than existing approaches and demonstrates superior performance compared to LoRA and full fine-tuning in a single fine-tuning experiment across 20 tasks. Additionally, our merging experiments show that sparse adapters not only retain strong performance on held-in tasks but also maintain competitive held-out generalization. While full-finetuning merging methods lead to performance degradation when scaled to 20 experts, sparse adapters prove to be more effective. Our approach enhances generalization on held-out tasks in comparison to traditional multitask training, though a performance gap persists when evaluated on held-in tasks. This study highlights the potential of sparse adapters as a scalable and efficient solution for constructing modular architectures, particularly as the number of tasks increases. These findings open avenues for future research aimed at closing the gap and further improving held-out performance.

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

## A    APPENDIX

**Computational Considerations of Model Mering Methods:**    As shown in Table 1, merging methods such as Task-Arithmetic (Ilharco et al., 2023), TIES (Yadav et al., 2023), and Breadcrumbs Davar (2024) require hyperparameter tuning to achieve optimal performance. We used the recommended hyperparameters for these methods. While sparse adapter only involves averaging weights when merging models, both TIES and Breadcrumbs require a TopK operation for each expert model to filter parameters, which is computationally expensive. The time complexity of the TopK operation is typically $O(n \log k)$, where $n$ is the number of elements in the input tensor, and $k$ is the number of top elements to retrieve. As the number of model parameters increases, the computational cost of this operation grows significantly.

**Importance of Recycling Subspace:**    The concept *Recycling Subspace* involves updating a selected subspace of parameters throughout the training process. Instead of discarding parameters like in pruning Lee et al. (2018); Arnob et al. (2021), we continuously recalculate their importance and adjust the subspace by adding or replacing parameters. This process allows for the reuse and recycling of parameters, improving model performance over time. After the first epoch, the subspace is kept fixed for the remainder of the fine-tuning. While  Lee et al. (2018) proposes the saliency criteria equation 1 for single-shot pruning, we find a significant improvement due to the iterative subspace update when tested on 5 tasks. Performance comparison is shown in Figure 4.

**Which Layer Should We Sparsify?**    We sparsify only the $QKV$ parameters in the attention layers of each transformer module. Although the output projection layer, $O$, can also be fine-tuned (Hu et al., 2021b), we find that fine-tuning only the $QKV$ parameters leads to better performance. We present empirical evidence in Figure 3, which compares performance across single-task and merged models, evaluating both held-in and held-out datasets after fine-tuning with a keep-ratio of $0.1$.

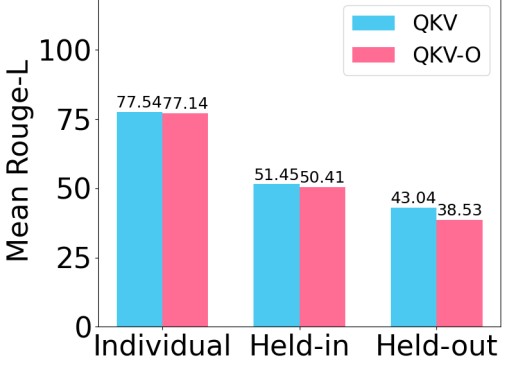
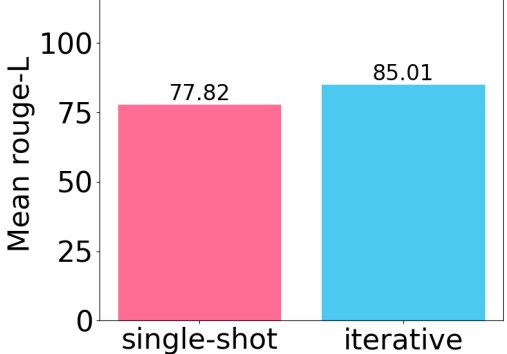

Figure 3: Performance of Sparse-Adapter ($kr = 0.1$) on training $QKV$ vs $QKV$-$O$ layers in Phi-3. Mean Rouge-L computed across 20 individual tasks and merged Performance for 20 held-in and 10 held-out tasks.

Figure 4:   Performance of Sparse-Adapter ($Kr$=0.1) on single-shot vs iterative update of the subspace. We compare the mean rouge-L performance of 5 individually trained tasks.

**Learning Rate Hyperparameter Tuning:**    We conduct a hyperparameter sweep to identify the optimal learning rate for fine-tuning the sparse-adapter, LoRA and FFT model. The mean performance presented in Figure 5 is evaluated across five fixed FLAN tasks, with learning rates varied at $1e^{-3}$, $1e^{-4}$, and $1e^{-6}$ to assess their impact on model performance.

**Tuning Block-Size Hyperparameter for Block-Sparse:**    We conduct an exploration of different block sizes, $B$ in block-sparse training to identify the optimal setting. As shown in Figure 6, we compare the performance of block-sparse training (with kr=0.1) across block sizes of 8, 16, and 32. Our results reveal that a block size of 16 delivers the best overall Rouge-L score for five individual tasks.

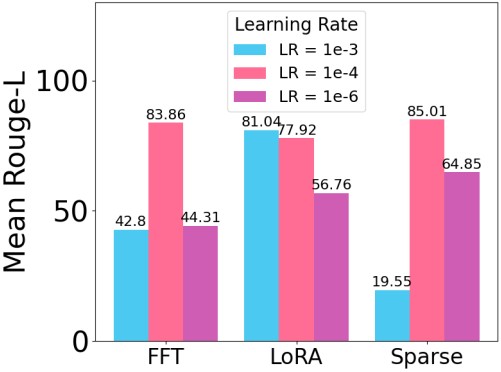

Figure 5: Performance of different methods under varying learning-rate. We compare the mean Rouge-L performance of 5 individually trained tasks to decide the best learning rate for each method.

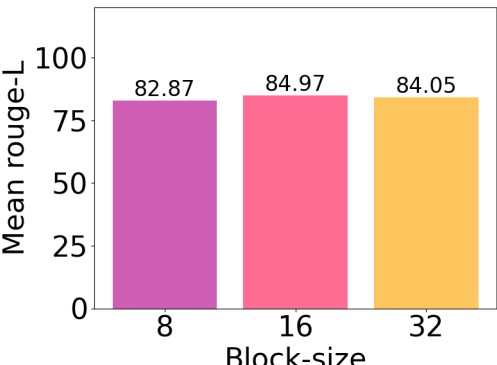

Figure 6: Performance of Block-Sparse-Adapter ($Kr$=0.1) on varying different block-size. We compare the mean Rouge-L performance of 5 individually trained tasks to decide the block-size.

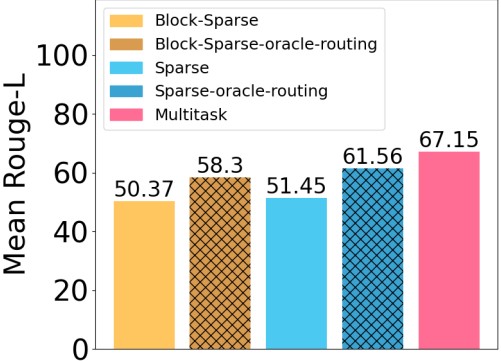

Figure 7: Performance Comparison of Oracle-Routing of the Sparse-Adapter with Multitask Performance over 20 Held-in tasks.

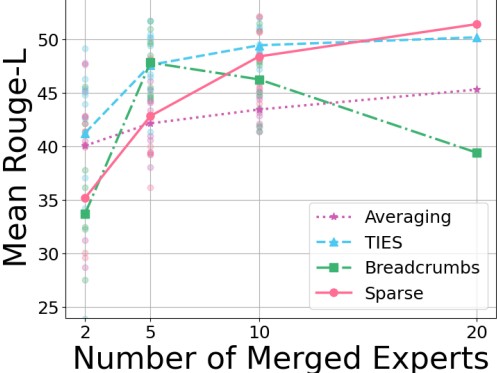

Figure 8: Performance comparison of different merging methods varying number of merged experts. For any number of experts, the merged model is evaluated on 20 tasks. The line is the averaged Rouge-L across all evaluation points.

**Performance Under Oracle Routing:** After merging the models, we compare the performance of individual tasks using Oracle routing in Figure 7, where we assume the task at hand is known, and find this further closes the gap with the multitask model. This approach is especially beneficial in situations with limited memory capacity, storing many fine-tuned experts is expensive. By using Oracle routing with sparse adapters, significant performance gains can be achieved, offering a more efficient solution. For practical implementation, this requires (1) task identification and (2) multiplying the merged weight in Equation 4 with task-mask: $\Delta W_m^* = m_i * \Delta W_m$. The performance improvement suggests, that the degradation in held-in performance is more related to parameter modifications made outside the sparse masks during merging than interference within the weight-overlap within the mask.

**Performance with Increasing Number of Experts:** In Figure 8, we evaluate the performance on 20 tasks for various merging methods as the number of merging experts, denoted as $N$, increases from 2 to 20, with the values $N = \{2, 5, 10, 20\}$. For each value of $N$, we conduct 10 trials and compute the mean performance across these trials. The figure presents both the performance variation and the mean performance for each $N$. We compare the performance of multiple merging methods on Full-finetuned models. Our findings show that as $N$ increases, the advantages of merging Sparse-Adapters become more evident, highlighting the benefits of sparsity as the number of experts grows. By $N = 20$, this method outperforms the other merging techniques.

