# OpenReview forum: "Exploring Sparse Adapters for Scalable Merging of Parameter Efficient Experts"
_ICLR.cc/2025/Workshop/MCDC — MCDC @ ICLR 2025_

### Official Review · Reviewer_RKzo · 2025-02-22

**Rating:** 4
**Confidence:** 4
**Fit:** 4

**Summary:**

This work explores the idea of sparse fine-tuning for performing Parameter Efficient Fine-tuning (PEFT) and improving model merging on NLP. The proposed framework uses the SNIP saliency score to periodically calibrate the mask during sparse adaptation. Experiments on NLP are carried out on established benchmarks, considering both held-in and held-out tasks merging performance.

**Reason For Giving A Higher Score:**

See Suggestions.

**Reason For Giving A Lower Score:**

N/A

**Strengths And Weaknesses:**

**Strengths**
- The proposed approach shows strong results, leaving no doubt in its benefits in terms of performance.
- Both held-in and held-out performance is assessed, which is interesting and a (sometimes) overlooked factor in model merging.
- The paper is clearly written and easy to follow. The pseudo-code (Algorithm 1) further enhances the understanding of the methodology.

------------------

**Major Weaknesses**

**W1.** The paper's contribution is very limited as Sparse Fine-tuning/Adaptation is not a novel concept [1,2]. The only difference I'm seeing is in the way the sparse mask gets calibrated (using a very well known Pruning-at-Initialization saliency score).

**W2.** I'm not clearly seeing the benefit in adopting sparse fine-tuning for memory efficiency (i.e. as a PEFT method), as it costs full memory at least in the beginning and I'm not quite seeing how it reduces the memory footprint after mask calibration. For instance, if a mask of a parameter has at least one non-zero entry, then it is unclear how it is possible to move that parameter on cpu and throw away the mask (as at least one element still requires to be updated). Also, the cost in memory of the masks seemingly is not taken into account, as well as the increased computation required by the element-wise multiplication with the masks.

**W3.** A comparison with other sparse fine-tuning approaches [1,2] is missing and would solidify the validity of this study.

[1] A. Panda, et al. "Lottery ticket adaptation: Mitigating destructive interference in llms." arXiv preprint arXiv:2406.16797 (2024).\
[2] L. Baohao Liao, et al. "Parameter-Efficient Fine-Tuning without Introducing New Latency." In ACL, 2023.

------------------

**Minor Things**
- I suggest to highlight better the contributions (eg. with bullet points) at the end of the Introduction.
- Also, it is not really clear whether a global or local masking approach is adopted when calibrating the masks.

**Suggestions:**

I would suggest the authors to thoroughly examine and report computational costs and memory efficiency analyses, as true sparse masks theoretical savings hardly translate to real improvements. Also, some minor clarifications would improve the treatment (eg. does the method consider a global or layer-wise ranking in the mask calibration logic?)

Finally, I would suggest to compare the proposed framework with other sparse fine-tuning approaches (see weaknesses).

---

### Official Review · Reviewer_9WoG · 2025-02-27

**Rating:** 7
**Confidence:** 3
**Fit:** 4

**Summary:**

The authors propose a novel, effective method to train sparse adapters for fine-tuning. They use an element-wise mask and a block-wise mask to make the weight updates sparse with an efficient training method. Additionally, there are experiments to analyse different merging techniques for various fine-tuning methods to showing the sparse adapters advantage on held-in tasks.

**Reason For Giving A Higher Score:**

The paper is well-written with clear motivation and comprehensive experimental results that illustrate the method's effectiveness. Naturally, since this is a PEFT method, it fits well with the theme of the workshop.

**Reason For Giving A Lower Score:**

Sparse adaptation is not an entirely new idea, even though the author's design of it may be novel. The method is comparable but does not improve on held-out tasks during merging.

**Strengths And Weaknesses:**

Strengths:
1. The training scheme for the sparse adapters is computationally efficient since the mask is only updated during the first epoch and only a small fraction of model weights are changed during fine-tuning, yet the procedure is highly effective in the single task setting as shown experimentally.
2. The writing is clear and well-structured, making the paper readable and easy to follow.
3. There is a comprehensive comparison with various merging methods to show the advantage of sparse adapters on held-in data when averaging them. This is probably due to the well-designed merging update of the sparse adapter to account for the parameter update overlaps.

Weaknesses:
1. There is a small ambiguity on the way the mask retains the parameters based on the saliency scores, if Top K or a threshold was used as both are mentioned but are different.
2. Regarding the memory requirements, the short discussion was appreciated, but perhaps a concrete comparison, along with an efficiency analysis would improve the paper.

**Suggestions:**

1. Having a table for the single task performance (Figure 1) would be more straightforward to understand your results.
2. The percentages on line 269 seem a bit confusing for me, I guess it is a 13.52%/11.01%/12.48% increase in performance as compared to the baseline but using the actual difference in rouge-L would be clearer for a reader.

Minor: Figure 2 is labelled with left and right instead of a and b, which is inconsistent with the text. Typo (equation) on line 160. Parenthesis citation on line 60.

---

### Official Review · Reviewer_YaPN · 2025-03-03

**Rating:** 6
**Confidence:** 4
**Fit:** 4

**Summary:**

The proposed method for training sparse adapters, and is demonstrated to be beneficial in the post-training merging process. However, the proposed merging method is a simple averaging, compared to more advanced model merging methods is missing.

**Reason For Giving A Higher Score:**

Refer to the strengths.

**Reason For Giving A Lower Score:**

Refer to the weaknesses.

**Strengths And Weaknesses:**

Strengths:

1. The paper is well-structured and clearly presented.
2. The paper demonstrates that sparse adapters outperform LoRA and full fine-tuning in certain scenarios, particularly in terms of parameter efficiency and merging properties.
3. The incorporation of saliency-based pruning to identify important weights is a well-motivated choice.

Weaknesses:

1. The proposed merging method is simple averaging. The paper lacks comparison with subspace-based model merging methods, such as Ties-Mering and TALL mask.
2. The paper lacks a strong theoretical foundation or analysis. While the empirical results are promising, the authors do not provide a rigorous theoretical justification for why sparse adapters outperform LoRA or full fine-tuning.
3. The paper lacks sufficient references to recent works on model merging.

**Suggestions:**

Refer to the weaknesses.

---

### Official Review · Reviewer_CSHa · 2025-03-05

**Rating:** 7
**Confidence:** 4
**Fit:** 5

**Summary:**

This paper presents a parameter-efficient fine-tuning method based on learned "sparse adapters" where first a sparse mask (either element-wise or block-wise) is learned using a saliency-based technique, then selected parameters (via sparse mask) are fine-tuned for the specific tasks separately. The author finds that using these sparse adapters outperforms the same size LoRA adapters and full fine-tuning.

Furthermore, the paper also experiments with merging the trained sparse adapters to evaluate on both held-in and held-out tasks. While merging sparse adapters outperforms all other compared merging methods (Averaging, Task-aritmetics, Ties, Breadcrumbs, and Lora merging) on held-in tasks, it falls slightly behind Ties merging on held-out tasks.

**Reason For Giving A Higher Score:**

I refer to the "strengths" of the paper I mentioned above as reasons for a high score.

**Reason For Giving A Lower Score:**

I refer to the "weaknesses" of the paper I mentioned above as reasons for a low score.

**Strengths And Weaknesses:**

Strengths:
1. The paper is well written, and experiments are carefully designed in terms of fair comparison for the proposed framework and the evaluation setting.
2. Although there are similar methods for sparse fine-tuning (see "weaknesses" below), this paper applies a saliency-based method to compute a sparse mask, unlike the previous work. Also, the authors show that updating the mask multiple times during the first epoch is useful.
3. In a single-task setting, the proposed sparse adapters lead to a better performance than LoRA adapters with the same size and interestingly also outperforms the full model fine-tuning.

Weaknesses:
1. I think the only major weakness of the paper is that there are similar sparse finetuning frameworks exist in the literature, as mentioned in the paper. There are certain differences, however, novelty of the proposed framework is limited.

**Suggestions:**

Although similar results have been reported where the sparse finetuning outperforms full model finetuning, it would be good to have a comprehensive analysis of this to validate that it is not a side effect of fine-tuning data size or hyperparameter selection.

---

### Decision · Program_Chairs · 2025-03-06

**Decision:**

Accept

**Comment:**

This paper expolores the idea of sparse finetuning as a PEFT method. Most of the reviewers agree that the paper well written and is a good fit for the workshop. Moreover, reviewers are mostly satisfied with the experiments in the paper. We believe that further clarifying the benefits in adopting sparse fine-tuning for memory efficiency would improve the paper. Specifically, how this sparse finetuning reduces the memory footprint furing training. Overall, the reviews are positive and hence we recommend acceptance.